# Clustering Redemption–Beyond the Impossibility of Kleinberg's Axioms

**Vincent Cohen-Addad**[*]
Sorbonne Universités,
UPMC Univ Paris 06,
CNRS, LIP6
vincent.cohen-addad@lip6.fr

**Varun Kanade**[†]
University of Oxford
varunk@cs.ox.ac.uk

**Frederik Mallmann-Trenn**[‡]
MIT
mallmann@mit.edu

## Abstract

Kleinberg [20] stated three axioms that any clustering procedure should satisfy and showed there is no clustering procedure that simultaneously satisfies all three. One of these, called the *consistency* axiom, requires that when the data is modified in a *helpful way*, i.e. if points in the same cluster are made more similar and those in different ones made less similar, the algorithm should output the same clustering. To circumvent this impossibility result, research has focused on considering clustering procedures that have a clustering quality measure (or a cost) and showing that a modification of Kleinberg's axioms that takes cost into account lead to feasible clustering procedures. In this work, we take a different approach, based on the observation that the *consistency* axiom fails to be satisfied when the "correct" number of clusters changes. We modify this axiom by making use of cost functions to determine the correct number of clusters, and require that consistency holds only if the number of clusters remains unchanged. We show that single linkage satisfies the modified axioms, and if the input is *well-clusterable*, some popular procedures such as $k$-means also satisfy the axioms, taking a step towards explaining the success of these objective functions for guiding the design of algorithms.

## 1   Introduction

In a highly influential paper, Kleinberg [20] showed that clustering is impossible in the following sense: there exists no clustering function, i.e. a function that takes a point-set and a pairwise *dis-similarity* function[4] defined on them as input, and outputs a partition of the point-set, that simultaneously fulfills three simple and "reasonable" axioms—scale invariance, richness and consistency. Scale invariance requires that scaling all the dis-similarities by the same positive number should not change the output partition. Richness requires that for any partition of the point-set, there should be a way to define pairwise dis-similarities such that the clustering function will produce said partition as output. Finally, consistency requires the following: if a clustering function outputs a certain partition of a point-set, given a certain dis-similarity function, then applying this clustering function to a transformed dis-similarity function that makes points within the same part more similar and points in different parts less similar, should yield the same partition.

While seemingly very natural in the context of clustering, the last of these axioms, *consistency*, is somewhat questionable as has been discussed by researchers over the years (see e.g. [30] and

---

[*]Ce projet a bénéficié d'une aide de l'État gérée par l'Agence Nationale de la Recherche au titre du Programme FOCAL portant la référence suivante : ANR-18-CE40-0004-01.

[†]This work was supported in part by the Alan Turing Institute through the EPSRC grant EP/N510129/1.

[‡]This work was supported in part by NSF Award Numbers CCF-1461559, CCF-0939370, and CCF-1810758.

[4]We use dis-similarity rather than distance, as for the most part we don't require the point-set and the associated dis-similarity function to form a metric space.

references therein). Consider a dataset with a natural clustering consisting of $k$ parts. Kleinberg's consistency axiom allows a transformation of the dis-similarity function by which one cluster may be subdivided into two subclusters, such that points in the same subcluster are very similar to each other, but sufficiently dis-similar to points from the other sub-cluster. The transformed instance may require a different partition: for a "good" clustering with $k$ parts, it may be more suitable to define one cluster for each of the two new subclusters and re-arrange the partition of the remaining points. Alternatively, one may ask what is the "right" number of clusters in the new instance? Since the original instance had $k$ clusters and since one of the clusters got subdivided into two subclusters, it may be more natural to ask for a clustering in $k+1$ clusters for this new instance. Unfortunately, this is not allowed by the consistency axiom: the clustering should remain the same. This scenario can indeed be formalized as shown in Section 4, even in the case where the original clustering into $k$ parts is very *well-clusterable*.

We are not the first to notice the problem with the consistency axiom as defined by Kleinberg, see e.g. [23, 2, 14]. This impossibility result has been contrasted by a large body of research that argues that relaxing the axioms by restating them with respect to cost functions (clustering quality measures) resolves the inconsistency [14]. For example, in the influential blog post [30], it is observed that the outcome of such a transformation can change the "natural" number of clusters.

Perhaps one of the main issue with Kleinberg's axioms is that they fail to explain why some of the classic clustering objectives, such as the $k$-means objective function (see Definition 1.1), give rise to very popular algorithms such as $k$-means++ and Lloyd's algorithm that are very successful in practice[5]. This suggests that the impossibility result arises from instances that are unrealistic and contrived.

A way to overcome this impossibility result is to look beyond the worst-case scenario. Motivated by the thesis that *"clustering is difficult only when it does not matter"* (see e.g. [19, 13, 17]), one can hope that classic objectives such as $k$-means would satisfy the axioms when the input is *well-clusterable*. Unfortunately, we show that $k$-means fails to satisfy Kleinberg's consistency axiom even when we restrict attention to *very well-clusterable* inputs, in fact even for types of instances for which the $k$-means++ algorithm has been proven to be efficient [21], and as a result one may expect $k$-means to be the "right" objective function to optimize. (See Section 4 for a formal statement of this.)

## 1.1 Our contributions

Our work aims at bridging the gap between real-world clustering scenarios and an axiomatic approach to understanding the theoretical foundations of clustering. We see the problem of clustering as a two-step procedure:

1. Determine the "natural" number $k$ of clusters in the dataset;
2. Find out the "best" clustering with $k$ clusters.

The question of choosing the "correct" number of clusters is a very relevant one in practice because several of the commonly used clustering algorithms take the number of clusters $k$ as a parameter (cf. [1, 26, 27, 18]) and would yield nonsensical clusters if $k$ was not carefully chosen. Despite this, theoretical work on choosing the number of clusters is quite limited compared to the vast theoretical work analyzing various clustering algorithms. An approach employed quite often is the so-called *elbow* method, which itself can be defined in different ways. A natural definition is as follows: Consider an objective function (a measure of quality) for clustering into $k$ parts, and define $\mathrm{OPT}_k$ to be the clustering that minimizes this objective function. According to the elbow method, the "natural" number of clusters is defined as the value $k^\star$ that maximizes the ratio $\mathrm{OPT}_{k-1}/\mathrm{OPT}_k$ for $k \in \{2,...,n-1\}$, where $n$ is the number of data points. $k^\star = 1$ and $k^\star = n$ are explicitly ruled out as they would lead to trivial clusterings.

The intuition behind this approach is that the maximum gain in information is obtained, precisely when finding $k^\star$ groups instead of $k^\star - 1$. There is diminishing information gain when allowing more clusters beyond $k^\star$. This approach is widely-used in practice e.g. [27] and has led to interesting theoretical models of real-world inputs. As an example, Ostrovsky et al. [24] define a "real-world" input with $k$ clusters as an instance for which the $k$-means objective satisfies $\mathrm{OPT}_{k-1}/\mathrm{OPT}_k > 1+\varepsilon$ for a sufficiently large $\varepsilon$. In turn such data models have been used in theoretical work to better understand the success of algorithms such as $k$-means++ [21, 10].

Taking inspiration from this approach, we return to Kleinberg's axioms and amend the consistency axiom to take into account the potential change in the optimal number of clusters. More precisely, we required that the consistency (i.e. the partition of the input point-set) is preserved in the transformed instance *only if* the "correct" number of clusters in the new instance is the same as that in the original instance.

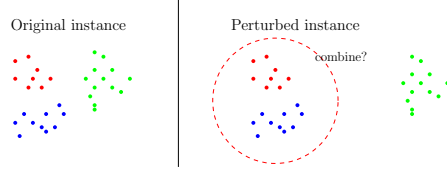

In order to meaningfully define the "correct" number of clusters, we need to include a *cost function*, from partitions to the positive reals, as part of the input. We define the "correct" number using the elbow method. We show that the new set of axioms is no longer inconsistent and some clustering algorithms, such as single-linkage, in fact satisfy these axioms. While in the worst-case, clustering algorithms based on the classic center-based clustering objectives such as $k$-means, $k$-median and $k$-center do not satisfy the axioms, we show that for *stable* clustering instances, these objective function now satisfy the axioms. We show that the notion of stable instance captures some interesting scenarios (see full version). Thus our axioms arguably model the process of clustering "relevant in practice" inputs, thus taking a step towards explaining the success of some popular objective functions.

**Stable clustering instances.** We define *well-clusterable* or *stable* clustering instance using the stability notion introduced by Bilu and Linial [16] in the context of center-based clustering. This notion was later considered in the context of clustering in several other works (see e.g. [9, 13, 15, 12]) and various (provable) algorithms have been designed for solving these types of instances. We consider the $\alpha$-proximity condition introduced by Awasthi et al. [11] which requires that the optimal clustering satisfies the following: Given a point in the $i$th optimal cluster, $\alpha$ times its distance to the center of cluster $i$ remains smaller than its distance to the centers of the other clusters. This notion generalizes the notion of stability as shown by [11].

In the full version we show that this notion arises for large ranges of parameters (for which our proofs hold) in different models such as the stochastic block model and mixture of Gaussians and for which these clustering approaches are used in practice. Our result on stable instances require that the cluster sizes are approximately equal; we observe that when using $k$-means in the context of e.g. Gaussian mixture models, roughly balanced clusters and a separation of centers ensures that the minimizing the $k$-means objective is roughly the same as finding maximum likelihood estimators for the centers.

## 1.2 Related Work

The authors of the prominent work Ben-David and Ackerman [14] were the first to defy Kleinberg's impossibility results. The authors focus on clustering quality measures (CQM), or cost functions, that assign clusterings a value. The authors interpret Kleinberg's axioms in terms of these quality measures and show that the interpretation of the axioms is consistent. This is similar to our approach but their definition of the consistency axiom differs: their notion of consistency asks for the value of a clustering to not increase after a perturbation of the inputs that bring points in the same cluster closer and pull points across different clusters apart.

As mentioned in the introduction, our consistency axiom requires that the optimal clustering remains the same after the same type of perturbation, if the optimal number of clusters remains the same. We believe that this is a stronger requirement that is of importance: when using a cost function for evaluating a $k$-clustering, we hope that the minimizer of the cost function (namely the optimal solution) is the underlying natural clustering. Hence, after a perturbation that does not increase distances between points of the same clusters and does not decrease distances between points in different clusters, we hope that the minimizer has remained the same if the natural number of clusters has remained $k$. The axiom proposed by Ben-David and Ackerman [14] does not enforce an optimal solution to remain optimal under the perturbation.

Ackerman [2] and Ackerman et al. [4] also contribute with a large set of axioms or properties that are suitable for clustering objective functions. In this paper we focus on the three original axioms introduced by Kleinberg. Thus, our approach aims at complementing their study of the axioms by replacing their consistency axiom with a stronger one. Also our approach differs slightly in the

following sense; we aim at defining reasonable axioms that explain why popular objective functions, such as $k$-means, are good ones (we refer the reader to [5, 6, 3] for further advantages of $k$-means and similar methods.

van Laarhoven and Marchiori [28] continue this line of research on quality measures and show that adding reasonable axioms leads to set of axioms which are not fulfilled by modularity, a fairly popular CQM. The authors of Puzicha et al. [25] explore properties of clustering objective functions for the setting where the number of clusters, $k$, is fixed. They propose a few natural axioms of clustering objective functions, and then focus on objective functions that arise by requiring functions to decompose into additive form.

Meilă [23] views clusterings as nodes of a lattice: there is an edge between to clustering $C$ and $C'$ if $C'$ can be obtained by splitting a cluster of $C$ into two parts. The authors give axioms for comparing clusterings and show inconsistency of those axioms. Ackerman et al. [5] considers clustering in the weighted setting where every point is assigned a real valued weight. The authors analyze of the influence of the weighted data on standard clustering algorithms. Ackerman et al. [6] analyze the robustness of clustering algorithms to the addition of points study the robustness of popular clustering methods. See Ackerman [2] for a thorough review on research on clustering properties. There has also been work focused on the single-linkage clustering algorithm and its characterization using a specific set of axioms including Kleinberg's axioms [31]. This has later been extended to more general families of linkage-based algorithms [3].

**Organization of the paper:** Section 1.2 introduces basic notions and notations. Section 2 describes and discusses our new axioms. Section 3 shows that single-linkage satisfies all of them, even in the worst-case scenario, while $k$-means and $k$-median satisfy the axioms when we restrict our attention to well-clusterable instances. Section 4 shows various impossibility results: $k$-means does not satisfy Kleinberg's axioms even for well-clusterable instances. In the worst-case, none of $k$-means and $k$-median satisfies all our refined axioms. The proofs can be found in the full version.

**Preliminaries** Let $[n]$ denote the set $\{1,...,n\}$. An input to a *clustering procedure* is $([n],d)$, where we $[n]$ is the point-set and $d : [n] \times [n] \to \mathbf{R}^+$ gives the pairwise distances between points in $[n]$ (we assume $d$ is always symmetric). We do not require $([n],d)$ to be a metric space, though all of our results continue to hold if this requirement is added. We denote by $\Pi[n]$ the set of all possible partitions of the set $[n]$; $\Pi^\star[n]$ denotes the set of non-trivial partitions of $[n]$, i.e. excluding the partitions consisting of exactly one part and the partition consisting of exactly $n$ parts. For a partition $\mathcal{P} \in \Pi^\star[n]$, we will denote by $|\mathcal{P}|$ the number of parts. We use OPT ($\mathrm{OPT}^o$, respectively) to denote the cost of the optimal solution under the perturbed metric (original metric, respectively).

**Definition 1.1** ($k$-Means)**.** *Let $([n],d)$ be a metric space, and $k$ a non-negative integer. The $k$-means problem asks for a subset $S$ of $[n]$, of cardinality at most $k$, that minimizes $\mathrm{cost}(S) = \sum_{x \in [n]} \min_{c \in S} d(x,c)^2$.*

In the $k$-median problem, the distances are not square while in the $k$-center problem, the sum is replaced by taking the maximum. In the following, we will sometimes refer to points of $[n]$ as *clients*. The clustering of $X$ *induced by* $S$ is the partition of $A$ into subsets $C = \{C_1,...C_k\}$ such that $C_i = \{x \in [n] \mid c_i = \mathrm{argmin}_{c \in S} d(x,c)\}$ (breaking ties arbitrarily). Similarly, given a partition of $X$ into $k$ parts $C = \{C_1,...C_k\}$, we define the centers *induced by* $C$ as the set of $\{\mathrm{centroid}(C_i) \mid C_i \in C\}$, where we slightly abuse notation by defining the centroid of set of point $X \subset [n]$ as the point $y$ of $X$ that minimizes $\sum_{x \in X} d(y,x)^2$ (a.k.a. the medoid). It is a well-known fact that $\mathrm{cost}(C)$ is minimized by the centers induced by $C$. Hence, we will refer to a solution to the $k$-means problem by a partition of the points in $k$ parts, or by a set of $k$ centers.

## 2 An Axiomatic Result

Kleinberg [20] introduced an axiomatic framework for clustering. Following Kleinberg, we define a *clustering procedure* to be a function $f$ that takes a pair $([n],d)$ of a point-set and an associated distance function, and outputs a partition $\mathcal{P}$ of $[n]$. This definition is purely combinatorial and in what follows we will modify it slightly to view clustering as an *optimization procedure*. Kleinberg [20] requires that any clustering procedure satisfy the following three axioms.

**Axiom 2.1** (Scale Invariance)**.** *For any input $([n],d)$ and any $\alpha > 0$, we have $f(([n],d)) = f(([n],\alpha \cdot d))$, where $\alpha \cdot d$ denotes an $\alpha$-scaling of the distance function $d$.*

**Axiom 2.2** (Richness). *For any $\mathcal{P} \in \Pi[n]$, there exists a $d_{\mathcal{P}} : [n] \times [n] \to \mathbf{R}^+$, such that $f(([n], d_{\mathcal{P}})) = \mathcal{P}$.*

The third of Kleinberg's axiom requires the notion of a $\mathcal{P}$-consistent transformation. For a partition $\mathcal{P} \in \Pi[n]$, a transformation $d'$ of $d$ is $\mathcal{P}$-consistent, if $d'(x,y) \leqslant d(x,y)$ if $x,y$ are in the same part in $\mathcal{P}$ and $d'(x,y) \geqslant d(x,y)$ if $x,y$ are in different parts in $\mathcal{P}$.

**Axiom 2.3** (Consistency). *If $f(([n],d)) = \mathcal{P}$ and if $d'$ is a $\mathcal{P}$-consistent transformation of $d$, then $f(([n],d')) = \mathcal{P}$.*

It is this last axiom that is an unnecessary restriction on clustering procedures. As discussed in the introduction, this restriction comes from the fact that the axiom enforces the number of clusters to remain the same, even after the perturbation of the input. Indeed, the *number* of clusters may have "changed" as a result of the distance transformation. In general choosing the correct number of a clusters is a fairly non-trivial problem. In order to do so, we assume that there is cost function associated with any partition $\mathcal{P} \in \Pi[n]$. To avoid trivial cases, we will only allow a clustering algorithm to output a non-trivial partition $\mathcal{P} \in \Pi^\star[n]$. Let $\Gamma : \Pi^\star[n] \to \mathbf{R}^+$ be a *cost function*. For any $k \in \{2,...,n-1\}$, define $\mathrm{OPT}_k^\Gamma := \min_{\substack{\mathcal{P} \in \Pi[n] \\ |\mathcal{P}|=k}} \Gamma(\mathcal{P})$.

For example, in the *so called* $k$-median clustering objective, $k$ data-points are chosen to be centers and each point is assigned to its closest center (with arbitrary tie-breaking) to arrive at a partition. Then the *cost* is simply given by adding up the distance of each data-point to its closest center.

We now present our refined consistency axiom. We consider a clustering procedure as a procedure that has as input $([n],d)$ as well as a cost function $\Gamma : \Pi^\star[n] \to \mathbf{R}^+$. The clustering procedure chooses the number of parts $k^\star$, by picking $k$ that maximizes the ratio $\mathrm{OPT}_{k-1}^\Gamma/\mathrm{OPT}_k^\Gamma$ and then outputs a partition $\mathcal{P}$ consisting of $k^\star$ parts that achieves the value $\mathrm{OPT}_{k^\star}^\Gamma$. We refer to such clustering procedures as clustering procedures with cost-function $\Gamma$ and denote the use $k^\star(([n],d),\Gamma)$ to denote the optimal value of $k^\star$ and $f(([n],d),\Gamma)$ to denote the partition output by the clustering procedure $f$ using the cost function $\Gamma$.

**Axiom 2.4** (Refined Consistency). *If $f$ is a clustering procedure with cost function $\Gamma$ and $f(([n],d),\Gamma) = \mathcal{P}$, then if $d'$ is $\mathcal{P}$-consistent, then either $k^\star(([n],d),\Gamma) \neq k^\star(([n],d'),\Gamma)$ or $f(([n],d),\Gamma) = f(([n],d'),\Gamma)$.*

What the above axiom states is that if a $\mathcal{P}$-consistent transformation does change data in a way that clearly changes the natural cluster structure, then it may output a different partition as the proposed clustering as long as the number of clusters has changed. However, if as per the objective function $\Gamma$, the "optimal" number of clusters has not changed, then the same partition $\mathcal{P}$ should be returned after a $\mathcal{P}$-consistent transformation. We refer to a clustering procedure using a cost function $\Gamma$ that satisfies Axioms 2.1, 2.2 and 2.4 as *admissible*. Section 3 establishes that unlike in Kleinberg's result which asks for clustering procedures satisfying Axioms 2.1, 2.2 and 2.3, we obtain a possibility theorem.

Several cost functions commonly used in practice have the effect of encouraging increasingly finer partitions. As a result, the number of parts, e.g. $k$ in $k$-means, has to be fixed to avoid achieving a trivial partition where each point is placed in its own clusters. On the other hand, it may be possible to imagine cost functions that encourage fewer clusters, e.g. if there's a cost to open a new center as in *facility location problems*. Based on these, it is possible to demand a stronger *consistency* axiom than the one state in Axiom 2.4. If $\mathcal{P} = f(([n],d),\Gamma)$ and $\mathcal{P}' = f(([n],d'),\Gamma)$, one may demand that if $k^\star(([n],d),\Gamma) < k^\star(([n],d'),\Gamma)$, then $\mathcal{P}'$ is a refinement of $\mathcal{P}$; likewise, if $k^\star(([n],d),\Gamma) > k^\star(([n],d'),\Gamma)$, one may demand that $\mathcal{P}'$ is a coarsening of $\mathcal{P}$. The former should be expected for cost functions encouraging finer partitions and the latter for cost functions encouraging fewer parts. Single linkage does have the property that a $\mathcal{P}$-consistent transformation can never decrease $k^\star$ and the resulting modified partition $\mathcal{P}'$ is a refinement of $\mathcal{P}$; however, we leave the formal analysis of this claim to the long version of this extended abstract.

## 3 Admissible Clustering Functions

### 3.1 Admissibility of Single Linkage

Single linkage is most often defined procedurally, rather than as an optimization problem. It is also commonly used as an algorithm for hierarchical clustering; however, it may equally well be viewed

as a *partition-based* clustering procedure. Formally, for a given $k$, the optimization problem that results in the single-linkage algorithm is the following: "Find the minimum weight spanning forest with exactly $k$ connected components (trees)".

As in any clustering procedure, the parameter $k$ is input to the algorithm. In order to *choose* the value of $k^*$, we look at the value of $k^*$ which maximizes the ration $\text{OPT}_k/\text{OPT}_{k+1}$ for $k \in \{1,...,n-1\}$. Note that this method of choosing $k^*$ does not allow $k^* = n$ nor $k^* = 1$.

**Proposition 3.1.** *Single linkage clustering is admissible.*

Proving scale invariance and richness is trivial. In order to prove refined consistency, we show that the optimal forest in the modified metric $d'$ with $k^*$ parts cannot use any edges that go between the trees in the forest obtained with $d$. We refer the reader to the appendix for the full proof.

**Remark 3.2.** *Actually, a stronger claim can be made where if $k^*$ changes, the new partition output by single linkage on $([n],d')$ will be a refinement of the partition output on $([n],d)$.*

## 3.2   Admissibility of $k$-Means

We now turn to a more formal definition of our "well-clusterable" instances.

**Definition 3.3** (Center proximity [11]). *We say that a metric space $([n],d)$ satisfies the $\alpha$-center proximity condition if the centers $\{c_1,...,c_k\}$ induced by the optimal clustering $\{C_1,...,C_k\}$ of $([n],d)$ with respect to the $k$-means cost satisfies that for all $i \neq j$ and $p \in C_i$, we have $d(p,c_j) \geqslant \alpha \cdot d(p,c_i)$.*

We further say that an instance is *$\delta$-balanced* if for all $i,j$, $|C_i| \leqslant (1+\delta)|C_j|$.

**Theorem 3.4.** *For any $\alpha > 5.3$, $\delta \leqslant 1/2$ and for any $\delta$-balanced instance satisfying the $\alpha$-center proximity, the $k$-means objective is an admissible cost function. Moreover, there exists a constant $c$ such that for $\delta \leqslant 1/2$ for any $\delta$-balanced instance satisfying the $\alpha$-center proximity with $\alpha \geqslant c$, the $k$-median objective is an admissible cost function.*

*Proof.* For simplicity, we assume that $\alpha = 6$ and $\delta \leqslant 1/2$, the general case is similar, the higher $\alpha$ the higher $\delta$ can be. We will only show the proof for $k$-means; the proof for $k$-median is analogous. The proof of all claims can be found in the appendix.

It is easy to see that the $k$-means objective function satisfies Axioms 2.1 and 2.2 (see also [2]). Hence, we only need to show that the $k$-means objective satisfies Axiom 2.3. We will make use of the following lemma mainly due to [12], and [11].

**Lemma 3.5** ([12]). *For any points $p \in C_i$ and $q \in C_j$ ($j \neq i$) in the optimal clustering of an $\alpha$-center proximity instance, we have $d(c_i,q) \geqslant \alpha(\alpha-1)d(c_i,p)/(\alpha+1)$, and $d(p,q) \geqslant (\alpha-1)\max\{d(p,c_i),d(q,c_j)\}$.*

We complement this lemma by the following observation:

**Claim 3.6.** *Given $p,q' \in C_i$ and $q \in C_j$, we have that*

$$d(c_i,q') \leqslant \frac{\alpha+1}{(\alpha-1)^2} d(p,q) \tag{1}$$

*and*

$$d(p,q') \leqslant \frac{2\alpha}{(\alpha-1)^2} d(p,q). \tag{2}$$

Consider an adversarial perturbation of the instance as prescribed by Axiom 2.3, namely a $C$-consistent transformation of $d$, where $C$ is the optimal $k$-means clustering of the original instance.

Assume towards contradiction that the optimal $k$-means clustering for the perturbed instance $\Gamma = \{\Gamma_1,...,\Gamma_k\}$, with centers $\gamma^* = \gamma_1^*,...,\gamma_k^*$, differs from the optimal $k$-means solution for the original instance $C = \{C_1,...,C_k\}$.

We claim that, assuming $\alpha > 2 + \sqrt{3}$ it must be that at least one of the clusters of $C$ contains no center of $\gamma_1^*,...,\gamma_k^*$. Indeed, if for each $C_i$ there exists a $\gamma_j^*$ that is in $C_i$, then the optimal clustering remains $\{C_1,...,C_k\}$ and so $\Gamma = C$. This follows from Claim 3.6: after the perturbation, each point of $C_i$ remains closer to points of $C_i$ than to any other point. Therefore, if there is a center $\gamma_i^*$ in each $C_j$, the optimal partitioning of the points remains $\Gamma$.

Thus, we assume that there is at least one cluster of $C$ that has no center of $\gamma_1^*,...,\gamma_k^*$.

In the following we aim at bounding $\text{OPT}_k, \text{OPT}_{k+1}, \text{OPT}_{k-1}$ which are the cost of the optimal solutions using $k, k+1, k-1$ centers in the perturbed instance.

We now consider the clusters of $C$ that contain no center in the solution induced by $\Gamma$. We also consider the centers $\{\gamma_1, ..., \gamma_t\} \subseteq \gamma^*$ induced by $\Gamma$ that are located in a cluster $C_i$ that also contains another center of $\gamma^*$.

Given a clustering $C'$ with centers $c'_1, c'_2, ..., c'_k$, we say a client $p$ is served by $c'_i$ if $d(p, c'_i) \leqslant d(p, c'_j)$ for all $j \geqslant i$ and $d(p, c'_i) < d(p, c'_j)$ for all $j < i$. For each $C_i$ that contains at least two centers of $\gamma^*$, let $A_i$ denote the clients served by all the centers of $\gamma^*$ located in $C_i$. We show:

**Claim 3.7.** *There exists $C_i$ that contains at least two centers of $\gamma^*$ such that $|A_i| \leqslant \frac{(\alpha+1)^2}{(\alpha-1)^2\alpha(\alpha-2)}|C_i|$.*

In the rest, we further analyze the structure of a cluster $C_i$ satisfying Claim 3.7. Let $\Delta_i = \max_{x \in A_i} \min_{\gamma_j \in \gamma^*} d(x, \gamma_j)^2$.

**Claim 3.8.** *We have that $OPT_{k-1} \leqslant OPT_k + \left( \left(\frac{(\alpha+1)}{(\alpha-1)^2}\right)^2 + \frac{(\alpha+1)^2(2\alpha-1)}{(\alpha-1)^4\alpha(\alpha-2)} \right)|C_i| \cdot \Delta_i$.*

**Claim 3.9.** *We have that $OPT_{k+1} \leqslant OPT_k - (1-\delta)\left(1 - \left(\frac{1}{\alpha-1}\right)^2\right)\left(\frac{\alpha-2}{\alpha}\right)^2|C_i|\Delta_i$.*

**Claim 3.10.** *We have that $OPT_k/OPT_{k+1} > OPT_{k-1}/OPT_k$.*

Claim 3.10 shows that if the perturbation creates a clustering $\Gamma$ different from $C$, then the natural value of $k$ has changed (namely $\text{OPT}_{k-1}/\text{OPT}_k$ is not the maximizer over all values of $k$). Hence, the axiom is satisfied.

It is easy to see that for larger $\delta$, a larger value of $\alpha$ allows to derive the proof. $\qquad\square$

## 4 Inadmissibility

In this section we prove two theorems showing inadmissibility of ubiquitous clustering functions.

**Theorem 4.1.** *$k$-means, $k$-median and $k$-center are not admissible w.r.t. our axioms.*

The following theorem shows that $k$-means, $k$-median remain inadmissible w.r.t. to Kleinberg's axiom even if $c$-cluster proximity is satisfied for any constant $c$. This is in contrast to Theorem 3.4 showing that $k$-means is admissible w.r.t. to our axioms if 6-cluster proximity is satisfied. Given that $k$-means of great importance in real-world settings, we believe that this is further evidence that our axioms are more suitable.

**Theorem 4.2.** *$k$-means, $k$-median are not admissible w.r.t. Kleinberg's axioms even when $c$-cluster proximity is satisfied for any constant $c$.*

### 4.1 Proof of Theorem 4.1

| $d(\cdot,\cdot)$ | $u_1$ | $u_2$ | $u_3$ | $u_4$ | $u \in L$ | $u \in R$ |
|---|---|---|---|---|---|---|
| $u_1$ | 0 | $\gamma-\varepsilon$ | $2\gamma-3\varepsilon$ | $2\gamma-3\varepsilon+1$ | 1 | $3\gamma-5\varepsilon$ |
| $u_2$ | | 0 | $\gamma-2\varepsilon$ | $\gamma-2\varepsilon+1$ | $\gamma-\varepsilon+1$ | $2\gamma-4\varepsilon$ |
| $u_3$ | | | 0 | 1 | $2\gamma-3\varepsilon+1$ | $\gamma+\varepsilon\ (\gamma-2\varepsilon)$ |
| $u_4$ | | | | 0 | $2\gamma-3\varepsilon+2$ | $\gamma$ |
| $v \in L, v \neq u$ | | | | | 2 | $3\gamma-5\varepsilon+1$ |
| $v \in R, v \neq u$ | | | | | | $\gamma$ |

Figure 1: Original instance and perturbed instance. Let $V$ be the set of points, with $|V| = n$. Assume $n$ is even, $\gamma = 1.5$ and $\varepsilon = 1/10$. Let $L$ and $R$ be two sets of size $(n-4)/2$ each. The perturbed instance is obtained by using the red value in brackets. Missing entries are given by symmetry and $d(u, v) = 0$ for $v = u$.

To see that $k$-center, $k$-median and $k$-means are not admissible, we will construct a distance function $d$ having $k = 2$ with a unique optimal clustering $C$. The instance is given by Figure 1 and we refer to Figure 2 for an illustration. Note that the distance function fulfills the triangle inequality albeit this is not required. The main idea behind the construction is that $u_2$ is, in the original instance, assigned to the cluster center $u_1$. In the perturbed instances, after decreasing the distance between $u_3$ and the nodes of $R$, we have that $u_3$ becomes the new center. As a consequence the node $u_2$ is now closer to that cluster than to the other cluster. Hence, the clustering changes. It remains to show that the optimal number $k^*$ of clusters remains 2 in the perturbed instance. Recall that, by definition, we

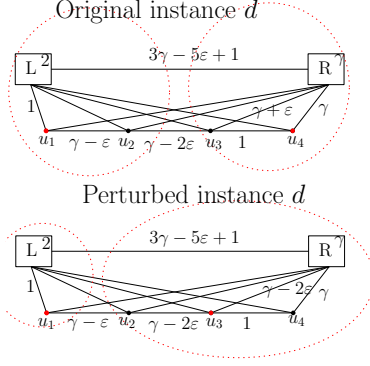

Original instance $d$

Perturbed instance $d$

Figure 2: An illustration of the instance given by Figure 1 that shows that $k$-center, $k$-means and $k$-median are not admissible w.r.t. to our axioms. The distance from $u_1$ to $u_2$ is $\gamma - \varepsilon$. The distance from $u_1$ to all of the nodes of $L$ is 1. The distance of a node $L$ to all other nodes of $L$ is 2 etc. The perturbed instances is obtained by decreasing the distance between $u_3$ and the nodes of $R$—all other distances remain unchanged. After decreasing those distances, the center shifts to $u_3$ causing $u_2$ to switch clusters and hence different clusterings. The red circles denote the optimal clusterings with the centers marked red.

exclude the cases $k^* = 1$ and $k^* = n$. We need to check that $\mathrm{OPT}_1/\mathrm{OPT}_2 > \mathrm{OPT}_k/\mathrm{OPT}_{k+1}$ for all $k \in \{2,3,...,n-1\}$.

$k$-**center.** Note that the optimal solution for $k=1$—in both the original instance and the perturbed instance—is to open a center at $u_2$. We get that $\mathrm{OPT}_1 = \mathrm{OPT}'_1 = 2\gamma - 4\varepsilon$. For the case that $k = 2$ we get for the optimal solution in the original instance (perturbed instance, respectively) consists of opening centers at $u_1$ and $u_4$ ($u_1$ and $u_3$, respectively). The results in a cost of $\mathrm{OPT}_2 = \gamma$ ($\gamma - 2\varepsilon$, respectively). Furthermore, note that for any $k < n$, $\mathrm{OPT}_k, \mathrm{OPT}'_k \geqslant 1$. As a result, we have $\mathrm{OPT}_k/\mathrm{OPT}_{k+1} \leqslant \mathrm{OPT}_2/\mathrm{OPT}_{k+1} \leqslant \gamma \leqslant 1.5 < \mathrm{OPT}_1/\mathrm{OPT}_2$ for $k \geqslant 2$; the same holds for $\mathrm{OPT}'$.

$k$-**median and $k$-means.** Consider $k$-median. Note that for $\mathrm{OPT}_1$ the cost is at least $(n-4)(2\gamma - 4\varepsilon)$ in the perturbed instance. Furthermore, $\mathrm{OPT}_2$ in the perturbed instance is $\mathrm{OPT}_2 \leqslant |L| \cdot 1 + |R| \cdot (\gamma - 2\varepsilon) + O(1)$. Hence $\mathrm{OPT}_1/\mathrm{OPT}_2 \approx 2$. We can easily verify that $\mathrm{OPT}_k/\mathrm{OPT}_{k+1} \leqslant 1.5 < \mathrm{OPT}_1/\mathrm{OPT}_2$ for all $k \geqslant 2$. The argument for $k$-means is along the same lines.

## 4.2 Proof of Theorem 4.2

For simplicity we consider an instance that satisfy 6-center proximity. The construction of the original and perturbed instance is given by Figure 3. Our construction for $k$-means and $k$-median is in fact the same and satisfies the triangle inequality before and after perturbation. We show that reducing the intra-cluster distance changes the optimal solution hence violating Kleinberg's axioms.

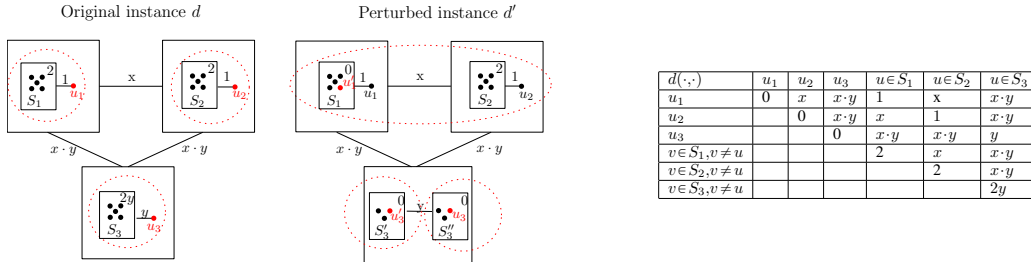

Figure 3: The two figures on the l.h.s. are an example of an instance that satisfies the 6-center proximity where $k$-means and $k$-median are not admissible w.r.t. to Kleinberg's axioms. The distance from $u_1$ to all nodes in $S_1$ (with $|S_1| = 5$) is 1, the distance for any node of $S_1$ to all other nodes of $S_1$ is 2. The distance from any node of $S_1 \cup \{u_1\}$ to any node of $S_2 \cup \{u_2\}$ is $x$ etc. The perturbed instance is generated as follows. First, the intra-distance between all nodes of $S_1$ reduces from 2 to 0. Second, the set $S_3 \cup \{u_3\}$ is partitioned into equal-sized sets $S'_3$ and $S''_3$. The intra-distance between nodes in both set reduces to 0 and the distance between a node of $S''_3$ and $S'_3$ reduces to $y$. All other distances remain unchanged. The red circles denote the optimal clusterings with the centers marked red. The table on the r.h.s. shows the original distance metric $d$. Missing entries are given by symmetry and $d(u,v) = 0$ for $v = u$.

We assume $x > \sqrt{5/2}$ and $y = 2x$. Note that the instance is 0-balanced and satisfies $x$-center proximity and also $x'$-center proximity for every $x' \leqslant x$, by definition. We require a few definitions. Let $u'_1$ be the red node of $S_1$ and let $u'_3$ be the red node of $S'_3$. The clustering $C_1$ induced by the centers $u_1$, $u_2$ and $u_3$ and simply assigning all other nodes to the closest node among $u_1$, $u_2$, and $u_3$. Let $C'_1$ be the

clustering induced by the centers $u_1$, $u_3$, $u'_3$. Let $C''_1$ be the clustering induced by the centers $u'_1$, $u_2$, $u_3$. Similarly, let $C_2$ be the cluster induced by the centers $u'_1$, $u_3$ and $u'_3$. Observe that the original metric space satisfies the $x$-center proximity definition. We will show that the optimal clustering $C_1$ in the original input and optimal clustering $C_2$ in the perturbed input are different. Hence Kleinberg's axioms are not fulfilled despite $x$-center proximity. For a clustering $C$ we use $\mathrm{cost}^o(C)$ and $\mathrm{cost}^p(C)$ to define the cost before and after perturbation.

$k$-**means.** Consider the original instance. We have that $\mathrm{OPT}^o_3 = \mathrm{cost}^o(C_1) = |S_1| \cdot 1^2 + |S_2| \cdot 1^2 + |S_3| \cdot y^2 = 10 + 5y^2$. Furthermore, $\mathrm{cost}^o(C_2) \geqslant \mathrm{cost}^o(C'_1) = 5 + 6x^2 + 4y^2$. We have that $\mathrm{OPT}^o_3 < \mathrm{cost}^o(C_2)$. Consider the perturbed instance. We have $\mathrm{cost}^p(C''_1) = 1^2 + |S_2| \cdot 1^2 + |S'_3| \cdot y^2 = 6 + 3y^2$ and $\mathrm{cost}^p(C''_1) \leqslant \mathrm{cost}^p(C_1)$. We have $\mathrm{OPT}_3 = \mathrm{cost}^p(C_2) = 1 + (|S_2| + 1) \cdot x^2 = 1 + 6 \cdot x^2$. Hence, the optimal clusterings in the original instance ($C_1$) and the perturbed instance ($C_2$) differ. An analogous reasoning yields the result for $k$-median.

## Footnotes

[5]Note that $k$-means++ and Lloyd's algorithm aim at minimizing the $k$-means objective; each step improves the quality of the solution w.r.t. $k$-means objective.

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
