[Reviews · NeurIPS 2018]

Reviewer 1



This paper extends Kleinberg’s work on the impossibility of clustering. That is, Kleinberg introduced 3 axioms that any clustering procedure should satisfy and then showed that it is impossible to satisfy all three simultaneously. This work suggests a refinement of Kleinberg’s 3rd axiom having to do with consistency. In the original axiom, if the clustering distance function is perturbed such that all within-cluster distances do not increase and all across cluster distances do not decrease, then the optimal clustering should be the same with respect to both the perturbed and unperturbed distance functions. The refined consistency axiom proposed in this work says that under the perturbed distance function, the optimal clustering may be different so long as the number of clusters in the optimal partition is different as well. The authors define the optimal number of clusters using the commonly used elbow heuristic. The remainder of the paper analyzes prevalent clustering algorithms and objectives like single linkage and k-center variants and shows under what conditions they do or do not satisfy Kleinberg’s first two axioms and refined consistency. The work has several strengths. The work positively contributes to the line of work on the impossibility of clustering, which helps to explain why clustering algorithms are effective in practice even when they are shown to be difficult to optimize--or impossible to optimize--in practice. The main new axiomatic ideas are simple, intuitive and crisp. For the most part, the writing makes the high-level concepts easy to understand. The theoretical results contribute to our understanding of common clustering objectives and are an important research thread to pursue. I think that researchers interested in a theoretical understanding of clustering will be interested in this work. The main issues with this paper seem to be easily corrected. In the proof of theorem 4.1 for the case of k-center, the authors claim that the optimal cost for k=1 cluster is 2\gamma - 4\epsilon, which they claim is achieved by opening a center at u_3, but I think this is incorrect; to me it seems that opening a center at u_3 incurs a cost of 2\gamma - 3\epsilon + 1. A better cost is achieved by opening a center at u_2 which incurs a cost of of 2\gamma - 4\epsilon (like what is claimed). Also given that in the example gamma == 1.5 and epsilon == 0.1, the final line in the proof, which claims that \gamma + epsilon <= 1.5 is incorrect. Perhaps what is meant is \gamma - \epsilon <= 1.5? Or perhaps that inequality should be eliminated? The writing of the proof of theorem 4.2 should be simplified. One thing that might help is a table showing what the various clusterings are, i.e., C1 = …, C1’ = …, C1’’ = …., C2 = …, etc. and their corresponding costs under the original and perturbed instances. Again, I think there is an issue in this proof: on line 341 there is a claim that OPT^o_2 < OPT^p(C2), which I think is incorrect based on the values of x and y given. However, this incorrect claim doesn’t invalidate the proof. Like in section 4, I think that the work would be strengthened by visual aids for a handful of the proofs in the supplement. There are a handful of grammatical errors and notational sloppiness in this paper which make understanding the technical arguments much more cumbersome. For example: in the first paragraph of 3.1 the optimal k is defined differently than the optimal k defined before axiom 2.4; on line 237 the authors write k \in {1, …, n-1} but then say that k cannot be 1; I think that k* and k^\star refer to the same thing--be consistent; after Theorem 3.4, the link to Axiom 2.3 should actually link to Axiom 2.4 (line 259); on line 274, it seems like \Gamma refers to a partition of the instances rather than a cost function--as it did in the beginning of the work; the supplement refers to Theorem 3.5 but I think this should be Lemma 3.5. There are other examples as well. There are some grammatical errors as well; for example, in the paragraph starting on line 138. Be sure to correct all of these mistakes. The introduction makes it seem like the theorems presented will be in terms of stable clusterings. Clearly connecting stability to \alpha-center proximity will improve the presentation. There is no conclusion. While I think it would improve the paper to include a conclusion--even if very short--I don’t find it absolutely necessary. As mentioned above, all of these issues can be corrected via a more diligent revision.

Reviewer 2



This is an interesting paper. The main point it makes, which it studies formally, is that Kleinberg’s impossibility result become more meaningful if we explicitly consider the number of clusters. In particular, the most essential and controversial axiom, consistency, should only hold if the underlying “consistent change” does not modify the number of clusters. It is then shown that, with the new formulation of consistency, Single Linkage satisfies the resulting set of axioms. Next, it is shown that under certain clusterability assumptions, k-means and related methods also satisfy the new set of axioms. Fundamentally, I really like this new direction, particularly the incorporation of the number of clusters into the axioms. I believe that it is promising and can lead to additional interesting findings. It is important to note that there have been many successful reformulations of Kleinberg’s axioms where they become consistent for Single Linkage. Perhaps the simplest is when we allow the number of clusters to be part of the input in our definition of a clustering function. The positive results of k-means require strong assumptions, both on cluster separation and balance. The practical implication of these combined assumptions isn’t clear (alpha>5.3 and cluster balance) - does it leave enough room for any ambiguous/realistic cluster structures? Nevertheless, I like the results, and elegant incorporation of the number of clusters into the formulation of consistency. This framework may also lead to additional interesting future work. Additional comments: - I wonder if the new formulation of consistency allows for any clearly unacceptable functions, which unnecessarily vary the number of clusters in order to avoid having to adhere to the core of the consistency requirement. - Perhaps it is worth incorporating a figure showing why consistency should consider the number of clusters? I realize that space it tight, but as this is the main message of the paper, it may be worth illustrating it with a figure, even though such figure appeared in previous work. Page 2, Step 2. Perhaps better written as “Find the best k-clustering” or “Find the best clustering with k clusters” Lines 114-116, it is not exactly a ‘perturbation’, since the change change in the dissimilarities can be significant. 129-131. There has been a couple of papers by Ackerman et al that aim specifically at understanding the type of properties that outline the advantages of k-means and similar methods, particularly: Margareta Ackerman, Shai Ben-David, Simina Branzei, and David Loker. Weighted Clustering. Proc. 26th AAAI Conference on Artificial Intelligence, 2012. Margareta Ackerman, Shai Ben-David, Sivan Sabato, and David Loker. Clustering Oligarchies. Proceedings of the Twelfth International Conference on Artificial Intelligence and Statistics (AISTATS), 2013.  - Jarrod Moore and Margareta Ackerman. Foundations of Perturbation Robust Clustering. IEEE International Conference on Data Mining (ICDM), 2016.  It is worth noting that the framework considered by Ackerman at al doesn’t necessarily propose k-means and similar methods are better than others, but rather seeks to understand under what conditions the advantages of k-means are relevant. This is most clearly seen in the first paper above, where it is proven that k-means (and similar methods) are sensitive to weight (or, data density). The other two papers show that k-means and related approaches are more robust to noise and outliers than other popular techniques, under clusterability assumptions. I see that some of these papers are already cited in your work, but are not contextualized to highlight that they do formally study the advantages of k-means and related methods. In that sense, focus on potential advantages of k-means is not a distinguishing element of the current work from the literature by Ackerman at el. What is different, however, is that they attempt to understand differences between algorithms for the purpose of determining which algorithms to use under different conditions, and your focus here is on the discover on consistent axioms of clustering that apply across diverse methods. Page 4: Perhaps it may be worthwhile to redefine notation to eliminate the repeated double bracketing. Line 198: Small comment: it’s -> its

Reviewer 3



The paper proposes a modification of Kleinberg's three axioms for clustering. In particular, instead of examining the property of a clustering function, the paper considers properties of clustering objectives. This is similar to the previous work of [Ben-David, Ackerman 09], but also taking into account the number of clusters and examining the optimum of a clustering objective instead of the cost. My comments are below: 1. In my opinion, the paper is of marginal significance, in comparison to existing work. 2. I have a question for the authors: in the negative cases (inadmissability section), optimal clustering for the instances are derived; how were you sure that these are indeed the optimum? For example, in Theorem 4.1, how can you justify that the optimal clustering of the original instance d is indeed an optimum (and the only optimum)?